# Olympic Infrastructure—Global Problems of Local Communities on the Example of Rio 2016, PyeongChang 2018, and Krakow 2023

**Bartosz Dendura**

Faculty of Architecture, Cracow University of Technology, 31-155 Kraków, Poland; bartosz.dendura@pk.edu.pl; Tel.: +48-12-374-2443

**Abstract:** Every potential Olympic Games organiser assumers that the games they organise will be special, that they will help to set new, high standards both in the organisation of sports events themselves, as well as in the planning and carrying out of projects accompanying the games. Since the document entitled Agenda 21 was approved by the UN at the Earth Summit in 1992, environmental protection has become the mainstay of the discussion among the Olympic family, and the problem of sustainable planning has become one of the main features of published reports. The conclusion of the conference coincided with the Lillehammer Winter Olympics of 1994, which were praised as the first green games to be held, under the motto "White Green Games". Four years prior, in Nagano, its organisers advertised the games with the slogan "Coexistence with nature". London 2012, with its motto "Inspire a Generation", highlighted the educational aspect of the events and the words "environment" and "sustainability" were on the lips of everyone everywhere in the context of every activity the organisers were undertaking. During Rio 2016, the motto was "Green Games for Blue Planet". The goal of the article is to investigate whether these mottoes have been actually reflected in the measures taken by authorities and encompassing the period between proclaiming them as Games organisers, during the sports event itself, and during the post-Games period. The author focused on the two most recent Games (Rio 2016 and PyeongChang 2018) in order to verify whether the ambitious assumptions that filled all manners of reports have been made a reality. The diagnosis presented in the article indicates that the pursuit of satisfying the ambitions of authorities and organisers is often at odds with the precepts and concept of sustainable development. The results of the study are an essential starting point for the discussion on developing a new formula—that of the European Games, whose third edition is to be organised in Krakow in 2023, and whose respective subject matter is identical to the problems that an Olympic Games organiser must face in terms of infrastructure planning.

**Keywords:** sustainable development; olympic legacy; architecture of mega-events; IOC; PyeongChang 2018; Rio 2016; Kraków 2023; Olympic Games; European Games

## 1. Introduction

### 1.1. Olympic Games as a Method of Improving Quality of Life

Modern Olympic Games have become a manifestation of the currently understood concept of the Olympics and have, since their reincarnation in 1896, evolved to become global events of immense cultural (sports), political and economic significance (particularly for some). As a rule, the Olympic Games are meant to be an occasion to promote human physical perfection and a healthy spirit, meant to inspire society to lead a healthy lifestyle, become more open to other cultures, religions and, in a broader sense, to improve life on Earth through the fundamentally democratic idea of sports for all [1].

This concept has its historical roots in attempts at countering elitist, professionalist and narrowly specialising tendencies, which have been increasingly often seen in modern sports history. However, along with increased interest in the games, the growth of the budgets of host cities and the scale of Olympic projects, in addition to the changes in global interests and problems that affect contemporary society, the IOC has altered the path that theoretically leads towards this same goal. Now, healthy and happy life is to be supported by investment in sustainable sports infrastructure, the engagement of the youth, and the propagation of the notion of global unity, of which the 40-day-long Games are meant to be a short episode (perhaps not even the most important one), during which the citizens of the world can demonstrate their capabilities.

*1.2. Sustainable Architecture and the Olympic Legacy in the Context of the Games.*

The scale of Olympic infrastructure which needs to house hundreds of thousands of guests over a short period of time, along with the interference it causes within the local ecosystem, are a significant design problem. Over the past three decades, many reports highlighting the role of the constructed Olympic legacy and sustainable development have appeared. These two notions, although closely tied, are not synonymous. Legacy refers to long-term benefits derived from the event, while sustainable development refers to the strategy and process of building a project, whose goal is to achieve positive outcomes while minimising the negative environmental, economic and societal impact of the project. The implementation of the precepts of sustainable design as early and as broadly as possible can significantly improve the overall economic effectiveness of buildings, including lowering their construction, occupancy and maintenance costs, as well as their impact on the environment and the health of residents, creating a valuable Olympic legacy. Despite mistakes made in the past (or thanks to prior experience), multi-billion-dollar projects can be beneficial to host cities. There has been an observable increase in investment in renewable energy sources, e.g., solar and wind-power, the use of biomass to both heat and cool, as well as the collection and reuse of surface runoff, or new waste management systems.

The true test for the organisers starts when the Olympic flame is passed on. Up to the 1980s, reports did not say much about Olympic legacy. Awareness of what was to become of the stadiums after the Games was low. Organisers focused on the event itself, on the effective supervision of the games, managing the crowds of athletes, officials and tourists so that each could safely arrive, be accommodated, and return home. However, in the last decade of the twentieth century, along with an increase in awareness of just how much humans interfere with the natural environment, a change in the manner of thinking occurred among the members of the Olympic family, as well as in the manner of thinking about the legacy of the Games. Representatives of the Committee often highlight their involvement in environmental protection [2]. The document entitled, Agenda 21 Sport for Sustainable development" [3], approved at the 1992 Earth Summit in Rio de Janeiro, which is a collection of the precepts of sustainable development for sports and the natural environment, can be considered proof of this. Matters of environmental protection were some of the leading topics during the Centennial Olympic Congress, which took place in Paris in 1994. The IOC wanted its flagship products, the summer and winter Games, to be perceived as an opportunity for kick-starting long-term positive change in society and its attitudes. These events were also seen as a chance to initiate positive environmental change and a rise in ecological awareness (provided that the events would be properly organised and managed). To this end, numerous international programmes and political campaigns have been initiated, such as the alliance with the United Nations Environmental Programme (UNEP) [4], which is meant to ensure the priority treatment of matters of ecology and sustainable development, resulting in a list of environmental requirements that have from then on been mandatory for host cities. Out of concern for the good of the planet, since the end of the twentieth century, Olympic Games have been organised under the motto "sustainability through sport". In 2003 the IOC Olympic Games Study Commission published a report in which the significance of Olympic legacy was strongly highlighted, seeing the need to provide host cities and their residents with long-term benefits as a priority [5].

Sustainable development is one of the key points of "Agenda 2020", a document that outlines the future of the International Olympic movement, the set of Olympic vales and good practices. The fact that the Olympic Charter—the fundamental set of rules to be followed by the Olympic Movement—was amended to include a provision discussing the promotion of a healthy Olympic legacy as late as in 2003, is all the more striking. The political acknowledgement of Olympic legacy has become possible thanks to the mutual influence of many social, cultural and economic processes. Directed largely by post-war global economic development and the associated ideology of consumerism, the Games have been transformed from events accompanying world's fairs at the start of the twentieth century, to modest and small-scale sports events akin to the ones in London in 1948, to a global media festival watched by billions of viewers [6]. This transformation has been necessary, if only for image-related reasons, for serious politics and the associated big business to remain able to act under the thin and delicate veneer of the mission that is associated with the organisation of Games under the sign of the International Olympic Committee. Just before the 2018 United Nations Climate Change Conference held in Katowice, Marie Sallois (the IOC's director of corporate development, brand and sustainability) said that "we now ask the cities what the Olympic Games can do for them instead of asking what they can do for the Olympic Games" [7].

## 2. Aim of the Study and Methods

Despite the subject of sustainable development and ecology being discussed by a large community of specialists and often being the subject of academic studies, it is still poorly understood by decision-makers and the organisers of large events. This article has three goals. First, the study was intended to verify to what degree had sustainable development been reflected in the measures taken by the organisers of the Rio de Janeiro 2016 and PyeongChang 2018 Olympic Games and what problems will the organisers of the Krakow 2023 European Games have to face. Second, the article summarises the last two editions of the Games in terms of whether their organisers had managed to implement the precepts of sustainable development. The initial assumptions of the organisers and the scale of environmental impact of their actions were compared. Third, the article is another phase in a series of studies whose objective is to determine the common elements of joint strategies adopted by Olympic Games organisers in the context of Agenda 2020 and the notion of the Olympic Legacy and which can be considered model solutions.

This study was based on bibliographic study of the literature. Three groups of literature were studied: official reports prepared by Olympic Games organisers and non-government organisations, academic articles and media reports which can be considered valuable because of the independence of their authors from organisers and governments.

There is no single applicable definition that can allow one to state whether a certain round of Olympic Games has been organised in a sustainable manner or not. This is why it is necessary to establish a common denominator that will allow us to perform the comparison. Based on 17 tenets described as The Sustainable Development Goals featured in the document entitled "Transforming our world: the 2030 Agenda for Sustainable Development" [8] and on the taxonomy described in Sustainable Built Environment and formulated in a partnership with the UNESCO Division for Science Policy and Sustainable Development [9], we can distinguish five main categories that enable the assessment of the efforts of organisers to create sustainable Olympic Games. These are:

1. Urban development—planning should start with the selection of the site, including striving to utilise existing infrastructure, supplemented with temporary accompanying buildings, to the fullest possible degree. The selection of a site for the Olympic village should take into account the protection of the local ecosystem or linking up with the existing road network, enabling easy material transport and, during later stages, access to public transport. New transport infrastructure should be based on public transport (e.g. underground railway, trains, trams, and electric vehicles), powered using renewable energy sources, and prioritise pedestrians and

cyclists, who should be provided with safe paths and pavements. The scale of the planned structures, as well as the construction sites themselves, should be reduced as much as possible.

2.  Environment enhancement—fresh water access should be safeguarded, as it is slowly becoming a strategic resource. Each and every structure alters local hydrologic conditions and any negative consequences should be minimised already during the design stage. In turn, supplying domestic water consumes immense amounts of energy needed to treat, pump, transport and process wastewater. This is why surface runoff or grey water (which is not suitable for drinking, but can be used to flush toilets or water green areas) are used increasingly often. Clean air should be a priority, with reductions in greenhouse emissions through the use of low-emission technologies and the planning of green areas that can protect and restore biodiversity.

3.  Sustainable building during the construction (or modernisation) of buildings should strive to use reusable or recycled materials. Materials should be locally sourced and their life extended as much as possible. Avoiding the use of energy derived from fossil fuels is also stressed. Achieving energy independence is possible through the use of renewable energy sources and improving the energy performance of buildings, turning them into net zero-energy buildings.

4.  Sustainable legacy—the cost of a building is perceived not only as the expenses associated with its construction, but its actual value also includes the cost of maintaining or remodelling it, its demolition and the processing of the resultant waste and rubble. This is why a sustainable building should be flexible, adaptable to new functions throughout its entire life-cycle, and the building materials used to construct it should be eco-friendly, easy to recycle or cheap and safe to process.

5.  Human skills—engaging people in all of a project's processes, enabling them to grow and providing them access to education at every stage of their lives, makes it possible to shape an informed society. This includes operators, infrastructure maintenance personnel (utilities, roads, parks, buildings), as well as future users. This helps to raise awareness, while highly qualified personnel can ensure the optimal performance and maintenance of buildings fitted with increasingly sophisticated management systems.

## 3. Analysis of the PyeongChang 2018 and Rio 2016 Games

### 3.1. PeyongChang 2018

#### 3.1.1. PyeongChang 2018 against the Political and Economic Situation of South Korea

South Korea, which is one of the world's leading economies, is still a rapidly developing country. Despite its good economic situation, the country has been wracked by radical political change. In 2016, after a corruption scandal and public demonstrations by millions of citizens, the parliament dismissed President Park Geun-hye. The elections held in May 2017 were won by opposition Democratic Party leader Moon Jae-in. However, the turmoil of recent years has exposed problems in South Korean governance, including shady ties between the state and big business, as well as the lack of appropriate institutional oversight that could prevent abuse of power by dominant individuals. The excessive concentration of power in the hands of one person is also a problem. The many postulates of the new public administration included, among others, promises of a greater focus on the environment, which had suffered greatly because of the previous administration, primarily focused on supporting business. Although smaller centres are actively implementing the assumptions of Agenda 21 at the local level [10], at the state level, Korea remains an environmentally unbalanced society, largely based on the pursuit of maximising economic growth and relying on vehicular transportation. Environmental protection policy is still lacking and is not conducive to maintaining an ecological balance [11].

3.1.2. PyeongChang 2017—Organisation of the Games.

The mountainous region of Gangwon is one of the reasons for which tourism is currently seen as a promising path of development for this area of Korea. Prior to the start of the Games, the central government selected an area of around 30 km$^2$, which covered parts of the provinces of PyeongChang, Gangneung and Jeongsean, assigning it for the construction of a future "Mecca of winter sports", planning to invest around 3.1 billion USD into new infrastructure [12].

During the pre-Olympic period, the organisers of the PyeongChang 2018 Games issued a number of reports on the progress of relevant development projects, as well as on how they were managing to implement the precepts of sustainability. The documents highlighted the efforts of planners in minimising the negative impact of the Games on the region and the environment, featured in an initiative entitled "New Horizons on Sustainability—Furthering Benefits to People and Nature". The initiative outlined five main areas of action, two of which focused on the environment, 2 on society and one on the economy [13]:

1.  Low Carbon Green Olympic Winter Games—a project with an additional subtitle "O2 Plus", it assumed the construction of certified, environmentally-friendly buildings, powered by energy from renewable sources, as well as effective waste management.
2.  Preserving and Environment—based on improving water quality, the restoration of 4 endangered species and the installation of devices meant to aid in managing various types of pollutants.
3.  Pride in Being a Mature 'Culture-Oriented Citizen'—a campaign meant to raise awareness among the region's citizens, encouraging them to expand their horizons and become open to culture. It included the activation of people with disabilities.
4.  PyeongChang on the Global Stage—Efforts meant to raise awareness about security and safety, as well as the precepts of sustainable development in accordance with international standards. This field included measures such as natural disaster preparedness training or in the ability of cooperating at a regional level.
5.  Healthy and Well-Rounded Life—an economic plan including the construction of sports and tourism infrastructure, assumed to have a positive effect on the region's popularity and, by extension, its economy.

While designing projects in individual centres, the organisers highlighted the meticulousness in their efforts to preserve the natural environment. One example of such measures listed by the organisers was their conduct during the construction of the Alpine sports centre. Initial assumptions featured skiing runs within an area designated as a genetic reserve forest, which is of great ecological value. Pressured by protests from the local community and the media, experts showed the organisers a different site. The committee also pledged to plant trees along the upper part of the runs and returning them to their original state from before the Games, in addition to covering an area twice the size of the one which was to be irreversibly damaged for the purposes of building Olympic infrastructure with newly planted trees.

3.1.3. PyeongChang—Were the Promises Fulfilled?

Although relatively little time has passed since the closing of the Games, certain conclusions can already be drawn.

Sustainable urban development—during any edition of the Games, transport is responsible for around 50% of greenhouse gas emissions. This problem is particularly observable in the case of winter games, when large groups of people need to be transported to mountain centres located at a considerable distance from the nearest city. The organisers of the Games in Korea estimated that around 1.7 million people would try to reach the areas of the Games' sports venues during competitions. This is why they placed a heavy emphasis on the development of environmentally-friendly transport. It was calculated that the Wonju-Gangneung Express Railroad would make it possible to reduce travel time to the area of the Olympic village from 5.5 hours to 1 hour. Because the express railroad emits

eight times less pollutants than traditionall-powered means of transport, this meant that the transport of every 0.5 million passengers produced a volume of greenhouse gases that was smaller by around 6500 m$^3$ [14]. In order to further improve these values, the organising committee provided additional electric and hybrid vehicles which transported athletes and members of the audience across Olympic venues. Large construction projects were carried out in the Gangneung region, where a landfill was converted into an Olympic Park, intended for conversion into a sports centre for the region's residents after the Games.

Environment enhancement—one of the pledges made by the organisers was to supply 100% of the energy needed to power the Games from renewable sources. To this end, twelve wind power plants were built in the region, while sports venues were fitted with solar panels and installations utilising geothermal energy. The departments of Gangwon province also pledged that at least 40% of the products they were to order were to be made from environmentally-friendly and recycled materials. However, the cutting down of 58,000 trees to make room for skiing slopes has cast a shadow on the environmental protection efforts of the organisers, with ecologists calling it an environmental catastrophe. The organisers only planted 181 plants as a replacement [15].

Sustainable building—the Olympic Winter Games PyeongChang 2018 were held in two separate clusters: snow events took place in the PyeongChang Mountain Cluster, and ice events were held in the Gangneung Coastal Cluster. A total of 12 venues and 14 training facilities were prepared. Among the twelve buildings that hosted the competitions, six were built from the ground up. As the organisers had promised, they were built from environmentally-friendly materials and received G-SEED certificates (eco-friendly Green construction certification that rates how buildings consume energy and reduce pollutants) [16]. Of the remaining buildings, four were equipped with energy reclamation installations intended to limit greenhouse gas emissions [17].

Sustainable legacy: Some months prior to the beginning of the Games, the IOC delegation inspecting the progress of the work noted the high maintenance costs of the buildings and that the lack of specific declarations as to their future could lead to the appearance of more "white elephants", which are expensive stadiums without a perspective of future use throughout the post-Olympic period [18]. Another element that was noted is that there were small chances for the region to become an important Asian winter sports centre. The temporary Olympic stadium, which served only to perform two opening and two closing ceremonies (both for the Olympic and Paralympic Games), was dismantled immediately after the conclusion of the Games. In the last months of 2019 there were media reports that new functions had already been planned for nine out of the twelve buildings, intended to sustain them in the post-Olympics period. All of the apartments in the Olympic village have been sold, while the media building has been converted into a training centre. Up until September 2019, the PyeongChang 2018 Legacy Foundation, which manages the Games' Olympic Legacy, was unable to reach a decision as to the fate of the Alpensa Sliding Centre, the Gangneung Hockey Centre and the Gangneung Oval, whose maintenance can cost the Gangwon province as much as 18 million USD up to 2022. These buildings are to be equipped with functions intended to supplement their existing offerings. However, in light of a lack of specific plans, it is difficult to say whether this decision can help the region avoid losses [19].

Human skills: Studies performed after the Games found that volunteers who worked to support the Games were very optimistic and satisfied. Among the main reasons affecting the positive reception of the Games, the respondents listed 1) a feeling of satisfaction derived from effective organisation (29.7%) 2) the satisfaction of and praise from foreign journalists and tourists (15.3%) 3) the positive message of the opening and closing ceremonies (12.2%) [20]. Cooperation between North and South Korea, promoting peace and harmony on the Korean Peninsula (at least during the Games) was also rated very positively. However, we must still wait to see the full effects of Olympic projects and the manner of their impact on residents, their growth and habit changes.

*3.2. Rio 2016*

3.2.1. Rio 2016 Seen against the Wider Events in Brazil.

During preparations for the Rio 2016 Olympic Games, Brazil was facing a political and economic crisis, whose scope and impact could not be foreseen earlier. As a result of this turmoil, the country was in a significantly worse economic situation on the day of the start of the Games than during its candidacy period. Despite these macro-economic factors, the government of Brazil constantly declared a willingness to tackle climate change, promising measures and investment towards a low-emission, resource-efficient and sustainable future compliant with the 2030 Agenda for Sustainable Development. These efforts were meant to include an improvement of water purity and easier access to water itself, extending territorial environmental protection, limiting poverty and greenhouse gas emissions [21]. At the same time, The National Institute for Space Research (Inpe) presented satellite images, confirming the alarming rate of the shrinking of the Amazon rainforest, to President Jair Bolsonaro. They clearly showed that the government administration had been assigning millions of hectares of the Amazon to agro-businesses, such as soy plantations, which have been pointed to as one of the major ecological threats to Brazil, particularly because of how global corporations, which are primarily motivated by profit rather than environmental protection, are interested in them. In addition, these plantations force and legitimise large infrastructural projects, as well as intense traffic. This sets in motion chains of events that lead to the destruction of natural habitats across large areas [22].

The state of the water in the Rio area was a separate matter. Since the 1950's they have been becoming increasingly polluted. This was caused by refineries, which operated without supervision in the middle of the century, as well as a dynamic migration of people that the area's technical infrastructure could not keep up with. In effect, almost half of the wastewater generated by the metropolis of nine million flows straight into Guanabara Bay, creating a dangerous mixture of bacterial and viral pollution [23].

Among the many negative, alarming reports, there are those which praise the efforts of the government in the field of environmental protection, while also noting that the polarisation of opinion among ecologists and agro-business representatives will make efforts towards sustainability more difficult [24].

3.2.2. Rio 2016—Organisation of the Games

In the past century, Brazil experienced an immense migration of people to urbanised areas and, as a result, over 80 % of its population currently lives in them. However, this rapid urbanisation has led to unsustainable urban development. The areas of cities that needed it the most have been left without resources, which have been redirected towards affluent districts, which are inaccessible to most. The efforts of successive governments aimed at changing this state of affairs often focused on attempts at removing illegal development, instead of searching for solutions that could improve the living conditions and comfort on-site [25,26].

In 2011, the Rio de Janeiro city hall published its first report on sustainable development [27]. During this time, the city's authorities were preparing for grand, historical events, such as the Rio +20 Climate Summit, the Football World Cup of 2014 and the Rio Olympic Games of 2016. The report noted with satisfaction that the city had already begun the process of transforming towards a more sustainable direction and expressed expectations that the Olympic dream, that symbolises these changes, can become a reality. The essential goals that the institution set before itself included:

1.　improving the quality of and access to public services;
2.　protecting natural resources and public spaces within the city limits, including water quality improvement;
3.　providing equal opportunities for the children and youth of Rio;
4.　creating a foundation for sustainable economic growth;

5.  integrating city areas in cultural and planning terms;
6.  combating poverty within the city limits, and
7.  placing Rio on the world map of cultural and political capitals.

The first report, entitled "Rio 2016 sustainability report", was published in September 2014. On the one hand, it highlighted the significance of current projects to the city's development, while trying to distance the authorities from actions that had sparked public protests, which included evicting families from areas were the projects were being built. The report stressed that a part of these projects, irrespective of the Games, had been planned for years and thus care should be taken when attributing a causal relationship between the Olympic and Paralympic Games organisers and the eviction of 738 families. The families evicted from the Vila Autodromo area, a site where work that was strictly tied with the Games was being done, could choose between financial compensation and being given a new apartment, located in Paraque Carioca. The document presented a sustainability management plan, which assumed measures compliant with the precepts defined in ISO 20121, namely: 1) responsibility in actions and decisions made in social, economic and environmental terms; 2) bringing together all stakeholders through the measures being undertaken, regardless of any factors that could lead to discrimination; 3) integrity based on ethical principles, compliant with international standards of conduct, and 4) transparency of action, backed by regular publications documenting results and consequences [28]. The report featured plans of the construction of new sports facilities and their accompanying infrastructure, as well as broad measures meant to improve quality of Guanabara Bay's waters, including wastewater treatment (in 2007 only 12 % of the wastewater flowing into the bay was treated), preventing floods, and improvements in collecting and processing waste.

### 3.2.3. Rio 2016—Were the Objectives Fulfilled?

Three years after the games, many people still ask the question whether there is a positive Olympic Legacy to speak of in the case of Rio. We can, with all certainty, say that not enough has been done to achieve the initial objectives. The organisational committee is still indebted to its suppliers, and the arrest of Carlos Nuzman, the head of the Rio 2016 Committee and the Brazilian Olympic Committee, under corruption charges in 2017, does little to improve the overall impression. The scrutiny continues to this day, with accusations jointly voiced by organisations from Brazil and France. Numerous articles stating that the assumptions featured in Rio's candidacy proposal, headed with the motto "Green Games for a Blue Planet", could not be achieved, were published already during the preparations for the Games. The National Office of Control in Brazil demonstrated that two years prior to the opening of the Games, when some of the projects had not even been started, the initially planned costs of several of them were already exceeded by 7%–112 %. It also became clear two years prior to the Games that the waters of Guanabara Bay could not be purified on time. Mayor Eduardo Paes has personally admitted that this objective was not fulfilled [29] and last minute efforts were necessary to prepare the sites for competition. Methods used such as eco-barriers across the 17 rivers flowing into the Bay, 12 specialized boats to find and remove trash and helicopters that patrolled the air, searching the water for large pieces of trash and debris. Although these efforts reduced the chances of collisions between watercraft and large trash items, it did little to reduce bacterial or viral counts in the water [28]. The Rio Games also showed how traffic can be damaging to the environment. Flying an estimated 28,500 athletes and staff to Brazil for the 2016 Summer Olympics in Rio generated more than 2000 kilotonnes (kt) of greenhouse gases (GHG), not to mention the 2500 kt of GHGs associated with bringing in about half a million spectators [30].

Sustainable urban development: The improvement of public transport in Rio is counted among the greatest benefits derived from the organisation of the 2016 Games. Work focused on developing three modes of transport: 1) bus rapid transit (BRT—rapid transit bus lines that travel along separate, dedicated carriageways), 2) the underground railway, 3) the light vehicle rail (VLT—tram lines, the first of which was opened two months prior to the start of the Games [31]. In the document entitled "Rio 2016—Legacy" published in 2018, its authors highlighted that because of the rapid transit bus lines

connecting distant housing developments, extended tram lines and the new underground railway line, the share of public transport rose from 18% to over 50% [31]. Furthermore, Rio has become the city with the largest urban bicycle path network in Latin America (over 400 km of bicycle paths) [32]. Philippe Bovy, an expert on the subject of transport during Olympic Games events, highlighted that the projects that had been completed over the six-year-long preparation period allowed the city to make up for 30 years of neglect in this area [32].

Environment enhancement: Contrary to public transport, whose improvement is counted among one of the organiser's successes, efforts towards environmental protection left much to be desired. Efforts made to clean up Guanabara Bay have been unsuccessful, as mentioned before. The promise to plant 24 million trees, intended to balance the carbon dioxide emission produced during the preparations for the Games, has also been left unfulfilled. Up to the spring of 2015, official representatives admitted that only 5.5 million trees had been planted [33]. The construction of a new golf course also inspired protests from ecologists. First, Rio de Janeiro already had had two golf courses used to host international competitions. The decision to build a completely new venue, funded by real estate developers in exchange for permissions to build luxury apartment buildings in the area, had been approved in the quiet confines of the city hall's offices while ignoring public consultation and without preparing environmental impact reports [33]. However, the official report highlighted that the organisers managed to reduce greenhouse gas emissions by 26%, cut down the amount of solid waste by 25% relative to original estimates and significantly reduce water consumption, while environmental protection services helped to preserve the endangered species present at the site of the projects [34].

Sustainable building—the organising committee pledged to implement strict LEED certification guidelines for every Olympic project. This meant limiting the consumption of natural resources, using renewable materials, cutting down on transport costs by selecting locally-sourced materials, as well as the reuse of materials procured through demolition [35]. The post-Games report indicated that The International Broadcast Centre and the Future Arena should be regarded as model Olympic Games venues. The first, with a floor area of 79,000 m$^2$, was designed so that all of its elements could be reused in other buildings after the Games. The second is composed of prefabricated elements, which were intended to be used to construct four schools in the city [36]. However, in November 2019, journalists reported that the building still stands and successive ideas for its reuse have been rejected due to the cost of the associated remodelling [37].

Sustainable legacy: In February 2017, images of closed and neglected sports venues were shown for the world to see. The Olympic Aquatics Stadium, which had been designed as temporary, was not dismantled, with Business Insider counting it among the 12 most expensive buildings in the world that remain unused [38]. The photographs were seen as the actual representation of the Olympic legacy in Rio de Janeiro. Because of a lack of funds, Deodoro Park was closed just a couple of months after its reopening and the famous Maracana stadium was photographed with missing seats and a sun-scoured pitch. In 2017 the roof of the Velodrome burned down, damaging a part of the track after a balloon fell on it. In May 2018, the building was repaired and reopened for use. The building was rebuilt in May 2018 and opened for further use. Official reports of the organising committee that studies Rio's Olympic Legacy do point to a number of successes. According to these documents, most of the sports facilities are frequently used, primarily by schools aiming to activate children. They also indicated that the renovation and construction of 700 km of new municipal water supply, sewerage, and telecommunications infrastructure have improved the comfort of living of Rio's residents [32].

Human skills, networks, and innovation: The organisers were praised for the opening ceremony of the Games, in which the subject of protecting the environment and efforts meant to address climate change had played the leading role. If we combine this with an audience of 3 billion, then the educational effect, which raises awareness of the dangers faced by the residents of earth on a global scale, appears to be quite a feat. In addition, the organisers secured the involvement of athletes in tree planting efforts, which only helped to reinforce this message [39]. 15,958 schools were involved in an education campaign, allowing over 8 million to children to learn about environmental protection

and sustainable development [40]. On the other hand, the plans to revitalise and develop some of the city's districts were criticised. The municipal authorities were accused of placing the interests of private investors above those of the public, which only deepened the socioeconomic divisions between residents. The forced resettlement of the residents of favelas, which stood in the way of municipal projects, was also poorly received by the residents. As a result, 60% of Brazilians expressed concerns that the Games would ultimately do more harm than good [39].

In January 2019, Brazil's Institute for Applied Economic Research (IPEA) published a report that demonstrated that Rio de Janeiro avoided economic recession thanks to organising the Games and that had it not been for them, the region's economic growth would have been lower by 5,1 % [40]. It also counted improvements to infrastructure and transport as positive, directly affecting the lives of 3 million residents.

## 4. The European Games

### 4.1. The New Formula of Multi-Disciplinary Sports Events

In less than 4 years, Krakow is to be the host of the third edition of the European Games. The new formula of multi-disciplinary sports events is modelled after the Asian and Pan American Games that have been hosted for years (the first games were organised in 1951 in both cases), as well as the African Games (since 1965). The inauguration of the edition took place in Baku in 2015, while the organisation of the 2019 edition was given to Minsk. Both host candidates were seen as controversial because of their internal political situation, their stances on human rights and their economic grey zones, mentioned in Amnesty International Report 2017/18—the State of the World's Human Rights [41]. The European Games, although smaller in scale than their global counterpart, still require careful planning and a long-term vision of the development of the host city/region, while the subject matter of the use of infrastructure is not much different to the one concerning the infrastructure built for the Olympic Games. The organisers of the Games in Baku (supported by specialists from the London 2012 organising committee) listed the media building as an example of this. It differed from the one built for the London Olympic Games only in the number of ports to which journalists could plug their computers in. The subject matter of the remaining infrastructure was exactly the same. As Tables 1 and 2 demonstrate, the number of persons who participated in the European Games corresponds to that of the participants of the Olympic Games of the 1970's and 80's. These were already large-scale events, requiring extensive budgets.

**Table 1.** Increase in the number of sports, events, and athletes during the Olympic Games over the last 40 years.

| City | Year | Sports | Events | Athletes |
|---|---|---|---|---|
| Moscow | 1980 | 21 | 203 | 5283 |
| Los Angele | 1984 | 21 | 221 | 6802 |
| Seoul | 1988 | 23 | 237 | 8473 |
| Barcelona | 1992 | 25 | 257 | 9368 |
| Atlanta | 1996 | 26 | 271 | 10,630 |
| Sydney [1] | 2000 | 28 | 300 | 10,960 |
| Athens | 2004 | 28 | 301 | 10,625 |
| Beijing | 2008 | 28 | 302 | 10,942 |
| London | 2012 | 26 | 302 | 10,768 |
| Rio de Janeiro | 2016 | 28 | 306 | 11,303 |

[1] In December 2000, during a committee session, it was agreed that the games should include 28 sports, 300 events and about 10,500 accredited athletes. Points 9 and 10 of the Agenda 2020 confirmed the number of athletes admitted, while allowing the number of disciplines to change while maintaining 310 competitors. These arrangements allowed to stop the uncontrolled growth of the competition as seen in the above list.

**Table 2.** Number of sports, events and athletes during the first two European Games.

| City | Year | Sports | Events | Athletes |
|------|------|--------|--------|----------|
| Baku | 2015 | 30 | 253 | 5898 |
| Minsk | 2019 | 15 | 200 | 4082 |

One significant difference between the European Games and the Olympic Games is their transparency during the host selection stage. The agreements with host cities are not publicly disclosed and the public is not informed of their provisions during negotiations. In the case of the games planned in Krakow, many residents were not even aware of the efforts of the municipal authorities, learning of the hosting of the games post factum from press reports. This is concerning because proper preparations already during the initial talks with the committee of the games are of key significance to the effective and sustainable hosting of the games, which has been demonstrated time and time again.

*4.2. The European Games—First Two Editions*

4.2.1. Baku 2015

In order to better understand the subject matter of the European Games, it is necessary to analyse the two first editions of the Games. Although the manners of the preparation of the Baku and Minsk Games were completely different, as were their budgets, their goal was the same. Both the president of Azerbaijan, Ilham Aliyev and the president of Belarus, Alexander Lukashenko tried to use international events to legitimise their antidemocratic regimes. These events were intended to improve their position, both domestically and internationally. This is why there are so few reports and public information available on these Games, and the journalists who had spoken unfavourably of the organisers and their actions were refused entry into the two host countries. Only a very small number of press reports, rarely translated into English, pointed towards irregularities and management errors. The government of Azerbaijan spared no expense to promote the country and change its image from a country typically associated with oil drilling and industry. The organisation of international events is a proven method of doing so (e.g. the examples of the Barcelona 2012 and Turin 2006 Games). Some of the stadiums were built from the ground up (e.g. the Olympic Stadium for 70,000 spectators, the Palace of Water Sports or the new Olympic Village). At the same time, the refusal to let foreign journalists into the country and preventing observers from reporting on current events in it that could possibly undermine the image of these events has cast a shadow over them [42]. Reports lack any official information about making efforts to protect the environment or the planned positive legacy—efforts intended to ensure that infrastructure could be utilised in the future. However, there is information about involving the youth in cooperation with Unicef [43] (which the organisation was criticised for by western media [44]), the popularisation of sport [45] and the first truly large-scale use of the Online Cloud, which made it possible to limit expenditure associated with information technology infrastructure and increased the effectiveness of management, becoming a model solution for future Games [46]. Thanks to the iVillage Platform, both athletes and officials received the latest information through electronic means, eliminating the need to print flyers every day. According to official documents, the European Games held in the capital of Azerbaijan cost around 1.12 billion USD. However, independent sources claimed that this amount had been undervalued (among other things, it does not include the cost of building the stadiums) with the actual cost of the Games being somewhere between 5–9 billion USD [47] (The Guardian listed this amount at 6.5 billion pounds [48]).

4.2.2. Minsk 2019

Four years later, the priorities of the Minsk European Games were different. President Lukashenko's enthusiasm for sport is well-known, and the Games were yet another occasion to

improve opinion of him among the residents of Belarus. Promotional materials stressed that "in the 21st century, Belarus has confidently declared itself as a sports power in the international arena" and "Thanks to the victories of Belarusian athletes, the image of the state is being formed, patriotism is being educated" [49]. Similarly as in the case of Azerbaijan, there are very few reports, with most press materials focusing on what has been achieved while ignoring sensitive subjects such as freedom of speech, human rights or environmental protection. The Games were organised largely using existing infrastructure. Only the Dynamo Stadium and the shooting complex in Uruchye were subjected to thorough renovation [50]. Contrary to the case of Baku, no new Olympic village had been built and athletes were housed in student dormitories, of which only one had been built from the ground up. The authorities managed to get the aid of over 8500 volunteers, who supported the organisation of the Games. Sources independent of the government stated that students had been forced to volunteer and had been reassigned en-masse to dormitories with lower living standards [51]. They had also been prohibited from contacting with the media [52]. Press materials also featured enigmatic information about changes to public transport, aimed at making it more sustainable, which included the purchase of 80 electric buses that travelled between stadiums [53]. However, the greatest difference can be observed in the budgets of the two Games. According to the Belarusian Ministry of Finance, the organisation of the Games cost around 260 million USD [54], which is more or less 5% of the amount Azerbaijan had spent on organising the Games.

### 4.3. The European Games—Krakow 2023

Krakow will be the first organiser of the games from within the borders of the European Union, and which will be scrutinised not only by Brussels, but also numerous NGOs, which mostly espouse a negative view of the games. Meanwhile, the subject of Olympic legacy does not enjoy much popularity, as events of this scale have never been organised in the country. In 2012, the final of the UEFA European Football Championship was jointly organised by Poland and Ukraine (none of the matches were played in Krakow). Krakow attempted to organise Winter Olympic Games twice: first in 2006, together with Zakopane (the application was rejected by IOC delegates), and in 2022 (Krakow retracted its application after a referendum in which its residents expressed their opposition to hosting the Games). If the organising committee of the games in Krakow wants to organise them in accordance with the precepts of sustainable development, it will have to solve several important problems;

Sustainable urban development—the application documents indicated that other large cities of the Lesser Poland Voivodship will take part in organising the games. In this situation, various stadiums can be located as far as 200 km away from each other (Krynica-Olkusz 193 km, Tarnów-Zakopane 179 km), which will generate the necessity of additional transport, which has been indicated as one of the main factors affecting carbon footprint in practically all reports, leading, as a consequence, to the deterioration of the environment. The existing network of public transport connections is slow and cannot compete with individual modes of transport (as demonstrated in Figure 1, the distance between Krynica and Olkusz can be travelled in 2,5 hours by car; the same trip can last between 12 and 16 hours by train and requires several transfers). Only cities along the A4 highway and the modernised railway line (the Tarnów→Krakow→Chrzanów link), that run along the East–West direction, can be considered to have proper circulation between them. The links with cities to the North and South of this axis require comprehensive remodelling.

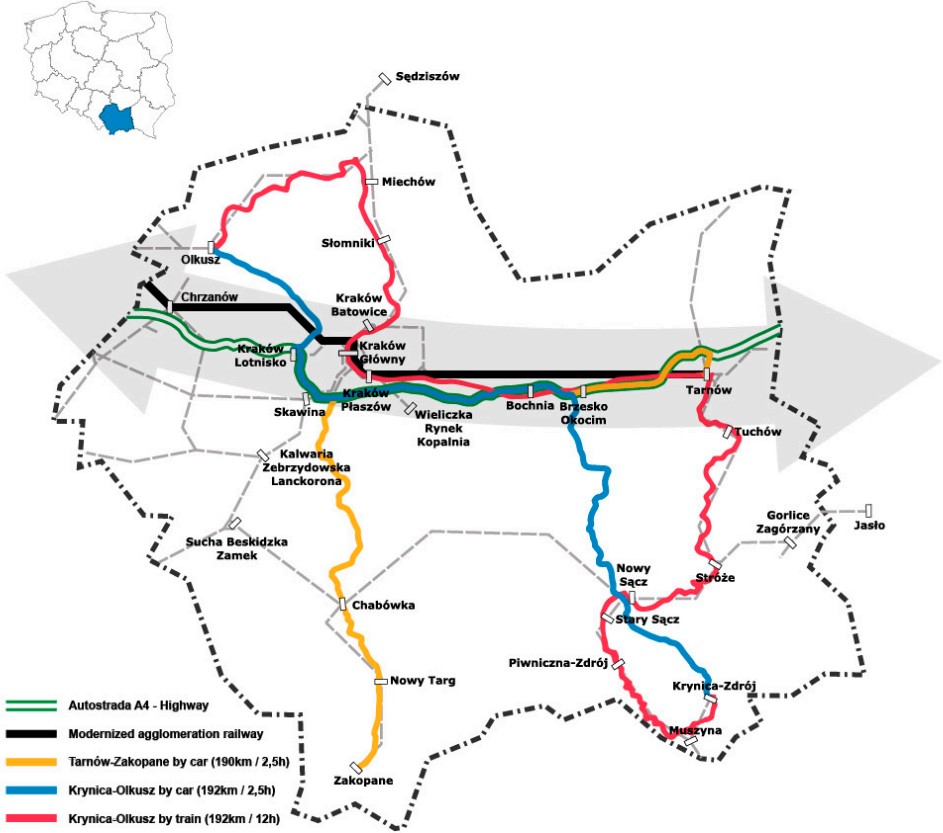

**Figure 1.** Circulatory connections between the cities of the Lesser Poland Voivodeship. The diagram shows travel time by car and by train between the cities that are at the greatest distance from one another.

Environment enhancement—another problem is the share of renewable energy in the energy mix. Poland, which has pledged to generate 15% of its energy from renewable sources by 2020, has reduced this share in its overall energy consumption for the second year straight, down to 10.9% (from 11.3% in 2018 and 11.7% in 2017) [55]. Energy is based on coal in the south of Poland in particular, with the vast majority of houses in rural areas generating heat using high-emission coal furnaces. The air of Lesser Poland's cities is counted among the most polluted in the European Union (the greatest number of days with emergency air pollution level indicators being exceeded). Although local authorities are undertaking various initiatives (on the 1st of September 2019 Krakow became the first city in Poland to completely ban using coal and wood as fuel), then the situation will not see any significant improvement without government programmes encouraging people to use and invest in renewable energy sources.

Sustainable building: Many reports referring to eco-friendly and sustainable building in Poland have recently appeared. In March 2019, the Polish Green Building Council (PLGBC) published its yearly report entitled "Polish certified green buildings in numbers—2019 analysis" [56], according to which Poland remains a leader in BREEAM, LEED, HQE, WELL, and DGNB certification in Central-Eastern Europe (51% of the region's certified buildings are located in Poland), Polish buildings have started to be certified during the construction stage through LEED Core & Shell certification and have obtained Existing Buildings certificates (this combination is much more often seen in the BREEAM certification system). A final WELL certificate in Poland has also been issued for the first time, as has a BREEAM certificate for a school buildings. Krakow came second in terms of the number of certified buildings. The presented statistics clearly demonstrate that the construction sector has become accustomed to the various certification methods and although this rarely pertains to public sector buildings, it does provide hope that this will change in the future. The accessibility of existing stadiums, their adaptation to current standards and their environmental impact are a separate matter. The organisers will have to

address the question of whether or not the modernisation of these venues is economically and socially feasible. This is why it is difficult to reliably estimate the need for stadiums and their environmental impact. Figure 2 shows the surface area of the city that the necessary Olympic Games infrastructure would occupy.

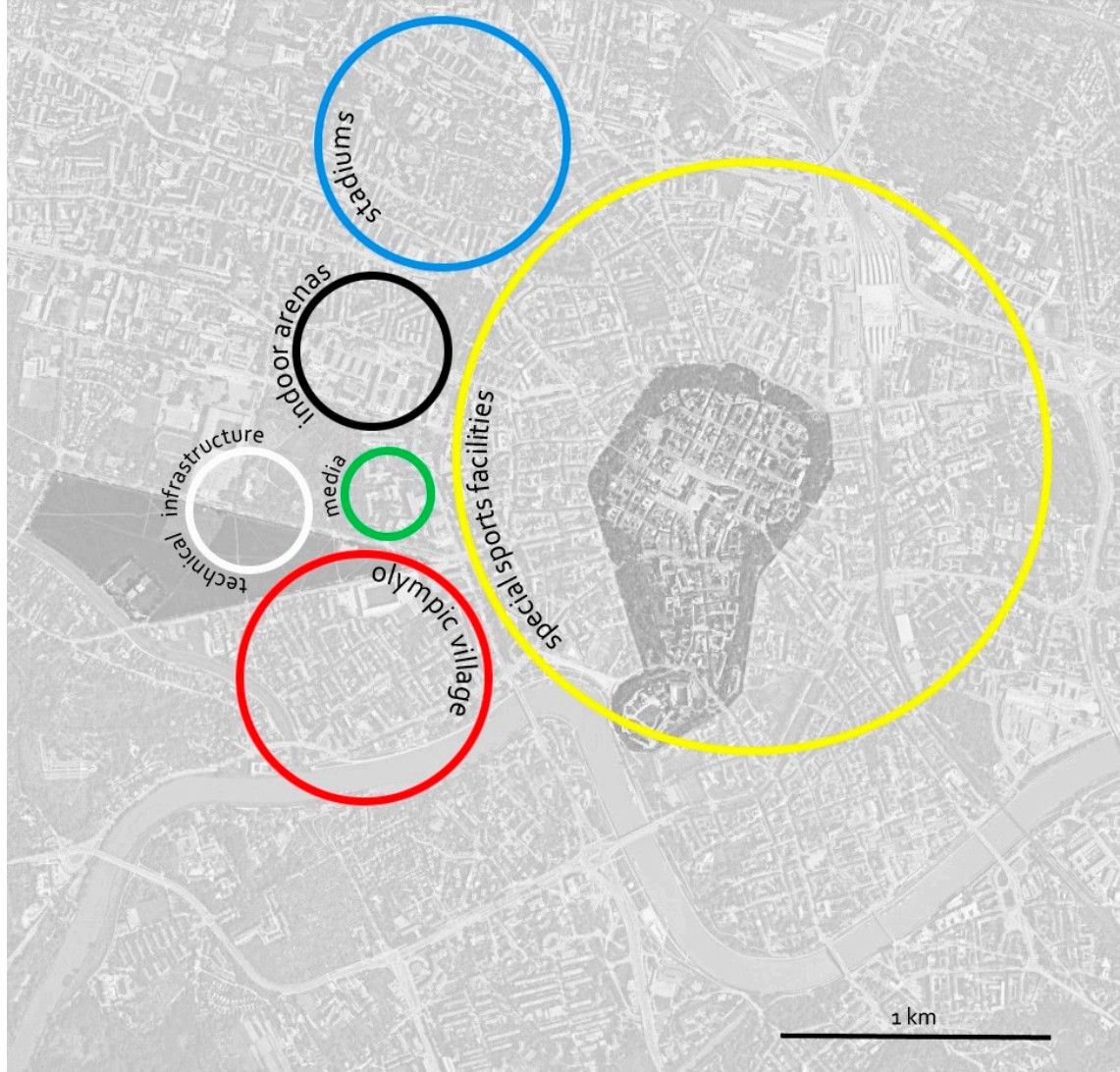

**Figure 2.** At the end of the 1990s, the IOC created guidelines according to which Organising Committees of the Olympic and Paralympic Games should provide min. 670 hectares of land for the Olympic park and Olympic village. This value includes stadiums (81 hectares); indoor arenas (32 hectares), press pavilions (11 hectares) and the entire technical infrastructure (20 hectares). A total of 445 hectares should be provided for special facilities such as rowing and kayaking track, swimming pool complex, shooting range and Olympic port; 82 hectares for the Olympic town, buildings of judges and observers as well as hotels and accommodation for the Olympic family.

Sustainable legacy—it is difficult to assess whether the planned European Games will leave a positive legacy without initial plans and designs. The only thing that can be said with certainty is that Poland has a shortage of high-level sports venues as many of them (particularly those in smaller cities) were built in the final decades of the twentieth century based on old technological solutions and environmentally-unfriendly construction materials. Their modernisation can only improve the general situation and enrich the offering of individual cities.

Human skills: Similarly as in the case of legacy, access to initial assumptions drafted by the Games organisers is required to assess the degree to and the fields within which the Games can leave a positive impact on residents. The idea of voluntary work has a strong following in Poland and bottom-up involvement in many projects is popular. A report published by the Centre for Public Opinion Research (CBOS) stated that 40% of respondents declared they had worked as volunteers in at least one type of organisation. This is the highest level of involvement with organisations in the past twenty years [57]. The participation rate of those who engaged in more than one type of activity also increased.

## 5. Results

Ever since I started investigating the subject of planning Olympic Games, with a particular focus on legacy planning and the impact of associated projects on the environment [58], I have been hearing that it is a problem that is simple and quite obvious. If it does not require further study and everyone already knows what they should be doing, then why does each organiser encounter so many problems? Further, even those Games which are considered to be model cases of the use of large-scale events in the process of urban development are still far from perfect. Searching for optimal solutions that guarantee sustainable Games is not easy (if it is possible at all). Cities and countries differ from each other and their environmental priorities are also different. They have different formulas of cooperation between local, state and NGO actors. Although the IOC tries to closely cooperate with organising committees, providing guidelines and aid based on the knowledge gained from previous Games, the messages sent to organising committees are often poorly interpreted.

In the Olympic Charter's second point it is written that IOC's mission is to promote a positive legacy from the Olympic Games to the host cities, regions and countries. In other words, organising committees are encouraged to invest in facilities that will serve local residents in the future, inspiring local communities to lead a healthy lifestyle, and that development standards set by public projects will become a reference point for subsequent investors.

However, the problem lies in the interpretation of these provisions by local authorities, which view organising the games as an opportunity to solve all of their region's ailments. It has been stressed numerous times that the Games are a trampoline allowing all the infrastructural projects that were planned for decades and for which no funds could be allotted to, to be carried out. However, the haste that accompanies such projects, along with the limited time available for planning, also necessitates non-standard solutions that are at odds with the precepts of sustainable development. Newspapers do not write about the carbon footprint, but an unfinished stadium would be clear as day. Reports say little of environmentally friendly transport based on renewable energy sources, but a traffic paralysis caused by an unfinished section of road/bypass/railroad tracks would be on the front page. The principle that the ends justify the means has guided the organisers of previous Games far too often. This is why it is so important to include the widest possible range of specialists in the earliest possible planning phase—specialists that could be able to formulate a long-term vision of the city's development and incorporate this vision into the short, episodic period of preparations for the Games.

A model example of the organisation of Games should be comprised of a pre-candidacy stage, during which a specially established Candidacy Committee would invite a planner, sociologist and economist with whom it would outline the framework of the event and incorporate it into the process of the development of the city. The perspective should encompass at least the next 30 years and include the short period of preparing for the games and an effective transition into the post-Games phase. Afterwards (expressing the desire and willingness of the majority of voters backed by appropriate public opinion polls) it can submit the city's application to host the Olympic Games, while simultaneously formulating guidelines for future projects in cooperation with designers. When a given city is selected, a plan of the construction of appropriate facilities should be prepared based on a prior analysis. If baseball is not a popular sport in a country, then it will probably remain unpopular there thirty years later, which is why the construction of a temporary facility for the sport is justified. However, if a sport like ice skating is enjoying widespread interest, but the country lacks a facility that would

meet requirements necessary to host an international-level competition, then it can prove beneficial to build it, not only in the financial sense, but also in the social one. Thanks to specialist consultations and appropriate preparation, such a facility could be designed on a well-prepared site, using local materials and built to a scale that is adapted to the actual needs of residents—in other words, in a sustainable manner.

In this study, I wanted to answer the question whether sustainable development and the pro-environmental promises featured in documents have been followed-up by appropriate measures taken by Games organisers. Today, when most environmental reports stress the deterioration of the eco-system caused by man's activity, treating ecological matters as a priority might appear obvious. The fact that a large share of the reports is not fully subjective and their focus is aimed at previously outlined areas is also a significant hindrance to effective study. Official documents of the IOC and the various organising committees stress the improvement of public transport, the construction of new parks and sports arenas that serve residents, as well as increases in the popularity of host regions, leading to greater profits derived from tourism.

**Funding:** This research received no external funding.

**Conflicts of Interest:** The authors declare no conflict of interest.

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
