# Peer review of "Olympic Infrastructure—Global Problems of Local Communities on the Example of Rio 2016, PyeongChang 2018, and Krakow 2023"

_sustainability, doi:10.3390/su12010141_

Round 1

Reviewer 1 Report

The work maintains, in general terms, an adequate scientific level and its object of study I value it as appropriate, in line with the thematic and title of the publication. I also consider the originality of this proposal to be a strong point, both in terms of the subject matter and the hypotheses under study.

The measured use of bibliographic citations is appreciated, focusing in general on sufficiently contrasted and relevant works. The structure followed conforms to the usual points and the suggested order for a scientific article and, with the points to be corrected below detailed, the work has an adequate wording and syntax, respecting the grammatical correction rules

Questions of method and content:

This is a study that addresses an issue of current relevance how the Olympic Games are, their new socioeconomic dimension and impact relating it to the principles of sustainability.

The justification for the case study is very appropriate. The goal of the article is to investigate whether these mottoes have been actually reflected in the measures taken by authorities and encompassing the period between proclaiming them as Games organisers, during the sport´s event itself, and during the post-Games period. The author focused on the two most recent Games (Rio 24 2016 and Pyeongchang 2018).

The results are consistent with the analysis and conclude with the need for sustainable development and the pro-environmental promises featured in documents to be followed-up by appropriates measures taken by Games organisers.

Author Response

Thank you for your review. I hope that the changes in the article introduced after reading other Reviewers’ comments won’t impact your initial appraisal.

Reviewer 2 Report

Introduction is much too long. It should be more succinct and present the central thesis for the reader. As it stands, much of what should be in this paragraph is found at the beginning of the second section of the study.

In terms of Rio (2016) and PyeongChang (2018), the author asserts that too little time has elapsed in order to make proper assessments at this time. As such, would it be beneficial to look at Sochi (2014)?

Author Response

Thank you for your review. I have shortened the Introduction according to your suggestion. The goal and methods of the research have been clarified, and the conclusions of the Rio (2016) and PyeongChang (2018) Olympic Games systematized.

In the analysis of the European Games, a short description of the first two editions in Baku (2015) and Minsk (2019) has been added. Due to the uniqueness of the Sochi 2014 Olympic Games, its catastrophic impact on the environment, no real control of the costs, but also a limited trust in the organizers’ reports, the Games has not been considered in the article. I hope that the changes introduced in the paper will make it more legible.

Reviewer 3 Report

The paper entitled "Olympic infrastructure—Global problems of local communities on the example of Rio 2016, PyeongChang 2018, and Krakow 2023" contains a thorough survey of the common elements of strategies employed by Olympic Games organisers.

Unfortunately, the part of the paper which has a relation with the subject matter of the journal is unsatisfactory. The provided analysis suffers from the formality and justification of their applicability to the problem at hand. Much more anylisis is needed.

Author Response

Thank you for your comments. The goal and methods of the research have been clarified, and the conclusions of the Rio (2016) and PyeongChang (2018) Olympic Games systematized and complemented with additional information and analysis.

In regard to the European Games section, a short analysis of the first two editions in Baku (2015) and Minsk (2019) has been added. Planned Games in Krakow (2023) have been analysed according to the same guidelines as for the Olympic Games in Rio and PyeongChang, which should give a better understanding of its specifics and issues.

I hope that the changes introduced in the paper will make it more legible and satisfactory.

Round 2

Reviewer 2 Report

The author has adequately addressed each of my previous comments.

Reviewer 3 Report

The author seem to have significantly improved the paper; thus it can now be published at its current form.